## REVIEW ARTICLE

# Causal contributors to tissue stiffness and clinical relevance in urology

Laura Martinez-Vidal [1,2✉], Valentina Murdica[2], Chiara Venegoni[2], Filippo Pederzoli [1,2], Marco Bandini[1,2], Andrea Necchi[1], Andrea Salonia[1,2] & Massimo Alfano [2]

Mechanomedicine is an emerging field focused on characterizing mechanical changes in cells and tissues coupled with a specific disease. Understanding the mechanical cues that drive disease progression, and whether tissue stiffening can precede disease development, is crucial in order to define new mechanical biomarkers to improve and develop diagnostic and prognostic tools. Classically known stromal regulators, such as fibroblasts, and more recently acknowledged factors such as the microbiome and extracellular vesicles, play a crucial role in modifications to the stroma and extracellular matrix (ECM). These modifications ultimately lead to an alteration of the mechanical properties (stiffness) of the tissue, contributing to disease onset and progression. We describe here classic and emerging mediators of ECM remodeling, and discuss state-of-the-art studies characterizing mechanical fingerprints of urological diseases, showing a general trend between increased tissue stiffness and severity of disease. Finally, we point to the clinical potential of tissue stiffness as a diagnostic and prognostic factor in the urological field, as well as a possible target for new innovative drugs.

The concept that malignancies have an increased consistency and rigidity compared to the surrounding healthy parenchyma is well known in medicine, as palpation of solid masses during physical examination has been the mainstay of tumor diagnosis for centuries. While this example represents an old-fashioned and unaware use of the mechanical properties of tissues for clinical purposes, Mechanics is emerging as a pivotal hallmark for several diseases and pathologies. The extracellular matrix (ECM) is a highly dynamic matrix that, far from being a passive filler, plays important regulatory functions in health and disease. Over the years, it has become clear that the ECM influences cell behavior not only by means of its chemical composition but also through its physical and mechanical properties[1,2]. Remarkably, single cancer cells seem to be softer and more deformable than their normal counterparts, probably to favor their movement and progression within the tumor microenvironment and beyond[3]. Indeed, mechanomedicine is an emerging multidisciplinary field focused on understanding how physical forces and cell and tissue mechanics regulate cell behavior, development, and tissue organization in physiological and pathological conditions[4]. Mechanomedicine, at the intersection between biology, medicine, physics, engineering, and material science, studies how specific diseases have particular mechanical fingerprints[5] (Fig. 1); this is true of cancer, fibrosis, aging, cardiovascular diseases, and chronic diseases as diabetes. Therefore, quantitative cell and tissue mechanobiology potentially offer the possibility to add a new class of disease hallmarks by identifying the changes in their physical properties.

Regarding living tissues, several mechanical parameters are of clinical relevance, among them stiffness is considered one of the most important and best described mechanical parameters of

[1] Vita-Salute San Raffaele University, Milan, Italy. [2] Division of Experimental Oncology/Unit of Urology, URI, IRCCS San Raffaele Hospital, Milan, Italy. ✉email: MartinezVidal.Laura@hsr.it

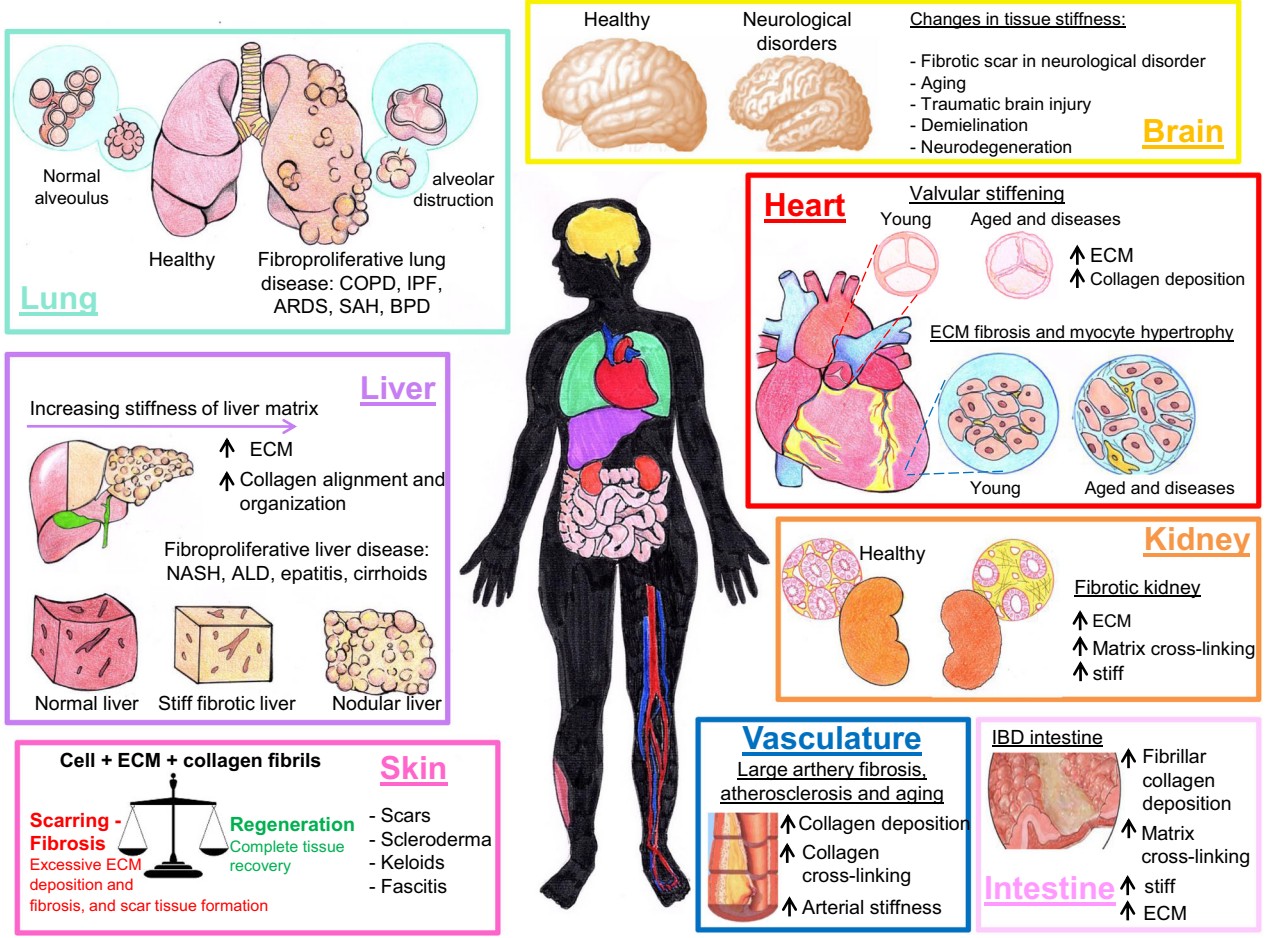

**Fig. 1 Schematic overview of the major diseases in which tissue stiffness is altered.** The hallmark of fibrosis is excessive extracellular matrix (ECM) synthesis and deposition that improve liver matrix remodeling and stiffening. In addition to change the amount of collagen deposited in the ECM, the alignment of the collagen fibrils is also contributing significantly to the alteration of the stiffness of tissues matrix. Increased matrix stiffness is not only a pathological consequence of fibrosis in traditional view but also recognized as a key driver in the pathological progression of fibrosis and cancer. Moreover, it is very likely that significant changes in cell and tissue mechanics contribute to age-related cognitive decline and deficits in memory formation which are accelerated and magnified in neurodegenerative states, such as Alzheimer's and Parkinson's disease. COPD chronic obstructive pulmonary disease, IPF idiopathic pulmonary fibrosis, ARDS acute respiratory distress syndrome, SAH pulmonary subarachnoid hemorrhage, BPD bronchopulmonary dysplasia, IBD inflammatory bowel disease, NASH non-alcoholic steatohepatitis, ALD alcoholic liver disease.

biological tissues in research[6,7] (Box 1), and thus our review will focus on stiffness. Pathological changes in tissue stiffness can be traced to altered amounts and/or function of its two fundamental constituents: cells (number and/or phenotype) and ECM (deposition and/or degradation). Knowledge of abnormal tissue states and the mechanical changes that occur in disease are important not only for diagnosis but also for tissue engineering and recapitulation of such diseases in in vitro models. These approaches are expected to generate novel therapeutic targets that allow for the restoration of healthy tissue mechanics and mechanotransduction responses[8,9]. Therefore, it is not surprising that mechano-based therapies that target increased tissue stiffness and associated cellular responses in cancer and fibrosis-mediated diseases are emerging clinically[10]. For instance, targeting of the renin-angiotensin system (RAS) by anti-RAS drugs has been shown to increase the anti-angiogenic effects of bevacizumab in liver metastases of colorectal cancer by reducing the ECM deposition and remodeling operated by metastasis-associated fibroblasts[11].

Herein, we aim to provide a general overview of the mechanobiology related to tissue stiffness, with a particular emphasis on the available data in benign and malignant urological conditions.

## Key contributors to a stiffer matrix
The ECM is a general term used to indicate the non-cellular tissue components that form an essential scaffold for cellular constituents. The structure of ECM differs in composition between tissues but is essentially made up of collagen fibers, proteoglycans, and multiadhesive matrix proteins that are secreted by different cell types[12]. The ECM is a dynamic component of tissues, where there is a constant feedback loop between the ECM and the cells within it: while cells produce and remodel the different ECM components, biochemical and biomechanical features of the ECM direct cell phenotypes. This equilibrium of the ECM turnover and crosstalk with the resident cells is crucial for tissue development and homeostasis.

**The cellular determinants of ECM remodeling.** Several cell types shape the ECM structure and composition. Among them, fibroblasts are the main producers of ECM components, including structural proteins (e.g., fibrillar and non-fibrillar collagens, elastin), adhesive proteins (e.g., laminin and fibronectin), and proteoglycans[12]. Moreover, they play a pivotal role in overseeing ECM quality and quantity, as they produce different enzymes involved in the maturation and catabolism of collagens, like lysyl

---

**Box 1 | Mechanics of materials: basic concepts and definitions**

Stiffness (rigidity): most used mechanical parameter for characterization of living tissues. It gives the resistance to undergoing (elastic) deformation in response to the application of a force, the property of being inflexible and hard to distort[216]. A stiff material, by definition, has a high Young's modulus (i.e., a considerable stress is needed for a minor deformation). The stiffness of a material is represented by the ratio between stress and strain, being the Young's modulus (E) the constant of proportionality between stress and strain, and one of the most common measures of intrinsic material stiffness. Units of Young's modulus are force unit per area, being Pascal (Pa) the most widely used.

Viscoelasticity: time-dependent anelastic behavior of materials. This means that the temporary response to a stimulus is delayed, and there is a loss of energy inside the material. Viscoelastic behavior normally occurs at different time scales (relaxation times) in the same material[217]. Viscoelasticity can be reported by Shear storage modulus (G′) and shear loss modulus (G″).

---

oxidase (LOX), which mediates collagen fibers crosslinking (thereby strengthening the ECM), and matrix-degrading enzymes, such as metalloproteinases[13]. Under physiological conditions, fibroblasts are generally quiescent, but they can be activated in response to a plethora of different mechanical and biochemical stimuli[14,15]. Chronic and deregulated fibroblast activation is at the basis of the altered ECM metabolism characterizing several diseases, including cancer. Chronically activated cancer-associated fibroblasts (CAFs) represent the main source of ECM production and remodeling within the tumor microenvironment (Fig. 2a), where they promote neoangiogenesis and epithelial to mesenchymal transformation (EMT)[16–19].

In addition to fibroblasts, macrophages express genes involved in ECM modulation, like transforming growth factor β induced protein and matrix metalloproteinase 9, which ultimately lead to the release of associated proteins and factors in the stromal environment[20–22]. For instance, a subpopulation of macrophages expressing the lymphatic vessel endothelial hyaluronan receptor 1 (Lyve-1) proteins has been shown to regulate ECM composition in the arterial wall and in the lung[21]. Moreover, macrophages can modulate fibroblast-mediated production of ECM, which contributes in turn to the regulation of fibrosis[23]. It is interesting to note that the mechanical status of the ECM can control the efficacy of TGF-beta activation by reparative fibroblastic cells and the release of traction-mediated latent TGF-β1 stored in the ECM[24]. Macrophages also play an important role in ECM remodeling in the tumor microenvironment. Tumor-associated macrophages (TAMs) may contribute to tumor progression by producing pro-inflammatory cytokines that will ultimately lead to apoptosis suppression and proliferation activation[25]. In addition, TAMs shape tumor ECM by secreting Matrix metalloproteases (MMPs) and matrix-associated proteins, and they organize collagen I into fibrillar bundles, as shown in a mouse model of colorectal cancer[18,26].

**ECM remodeling proteins.** LOX is an extracellular enzyme that catalyzes the crosslinking of collagen and elastin through oxidative deamination of lysine residues[27]. Reduced LOX activity in humans has been described in two X-linked recessively inherited disorders (e.g. Menkes disease and occipital horn syndrome)[28,29], while elevated LOX levels are clinically associated with increased systemic/organ fibrosis[30,31]. In cancer, LOX plays an important role within the tumor microenvironment since early stages of tumorigenesis[32] (Fig. 2b). By upregulation of LOX, CAFs increase collagen crosslinking, altering ECM topology, directionality and mechanical properties, ultimately leading to ECM stiffening[33], which promotes metastasis and infiltration of tumor-supporting immune cells[26]. A positive correlation between LOX expression and cell migration, invasion, EMT, and metastasis has been observed[26,34]. Furthermore, LOX overexpression is an indicator of poor patient prognosis, as seen in several tumors[33,35–38]. Thus, there is potential application of LOX inhibitors in clinical trials to facilitate permeability of drugs and infiltration by tumor-killing immune cells[39], taking into consideration that LOX inhibitors can only reduce the further crosslinking of collagen fibers, but not restore the already cross-linked ECM.

Matrix metalloproteases (MMPs) are secreted by tumor cells, CAFs, TAMs, and other stromal components (Fig. 2c). MMPs allow for ECM degradation, necessary for cancer cell invasion, and metastasis[18]. These proteases are implicated in almost all steps of metastasis, and an elevated level of MMPs is directly correlated with poor prognosis and a high risk of relapse[40,41]. Furthermore, MMPs release ECM-attached soluble growth factors and cytokines[42], including vascular endothelial growth factor (VEGF), which will ultimately promote neoangiogenesis, contributing to tumor growth and potential metastatic spreading[43,44]. For example, MMP-9 and MMP-2 cleave TGF-b, promoting tumor invasion, and angiogenesis[45–49]. Werb and colleagues pointed out the role of proteolysis as a mechanism of altering extracellular signaling[50]. For instance, they demonstrated that MMP-2 is a strong contributor to prostate carcinogenesis and that MMP-2 deficiency results in a reduction of immature blood vessel numbers[51]. Likewise, they found that MMP-9 has both pro- and anti-tumorigenic effects, depending on the environment and stage of cancer progression[50,51]. These discoveries fostered new paradigms about the role of MMPs, the microenvironment, inflammation in development and cancer and changed the way in which biomedical researchers view proteolysis, from ECM destruction to extracellular signal transduction[52]. After several clinical trials targeting MMP inhibition led to disappointing results about 20 years ago, probably due to an insufficient knowledge of the MMP interactome and of the pharmacology of the tested compounds, novel ongoing studies are testing MMPs ability to activate prodrugs or facilitate drug delivery[53,54].

## Tissue mechanics on disease: ECM deposition, topographic reconfiguration of the stroma and cell contribution to tissue stiffness

**ECM stiffness and malignant ECM modifications.** The ECM plays a crucial role in the classically defined hallmarks of cancer, from tumor onset, progression, and metastasis[34,35], and its biochemical and biophysical characteristics undergo constant remodeling[37,55]. During cancer and fibrotic diseases, dysregulated matrix synthesis and remodeling takes place. Cell components of the tumor microenvironment (cancer cells, CAFs and TAMs) modulate ECM through different activities. One of these modulations is the topographic reconfiguration of the stroma: ECM anisotropy (Box 2). The alignment of ECM fibers yields a rigid structure that contributes to tumor stiffness[11,19]. One example is the radial alignment of thick collagen bundles seen at the invasive front in breast cancer[20]. Linearization of collagen fibers has been associated with an increase of tissue stiffness, and both properties are related to the presence of neoplastic tissue and poor prognosis, while

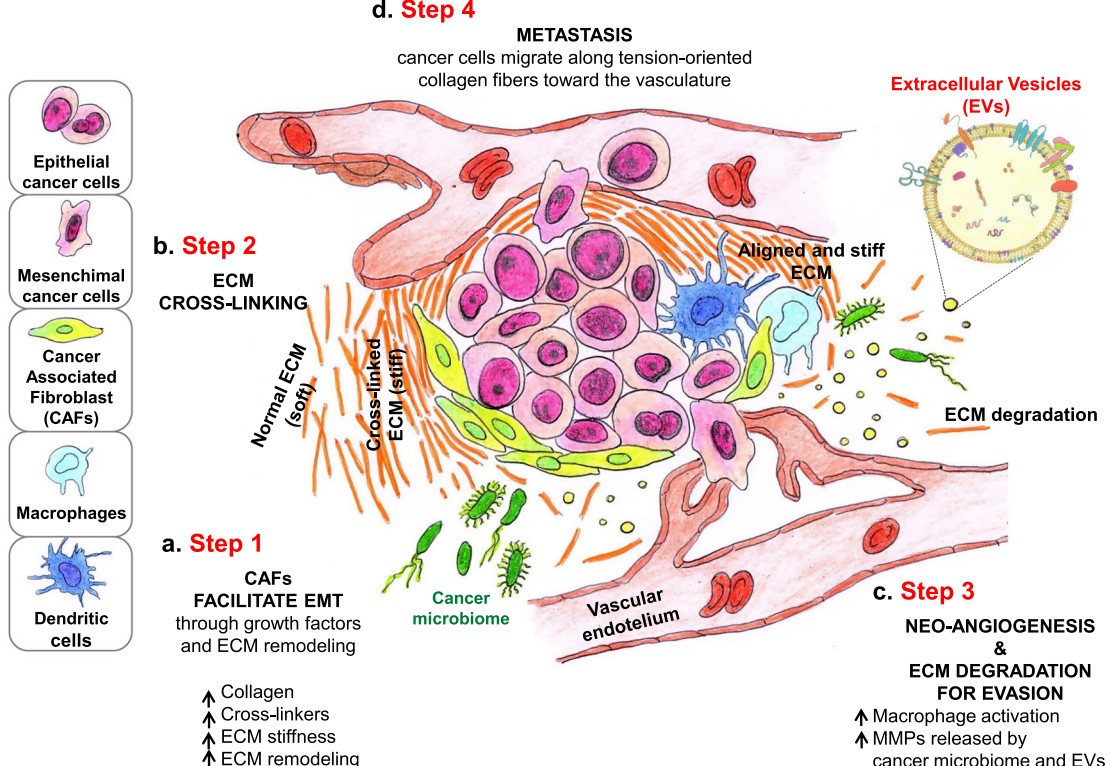

**Fig. 2 Extracellular matrix and tumor microenvironment.** Cell components of the TME (cancer cells, CAFs and TAMs) modulate ECM through different activities. One of these modulations is the topographic reconfiguration of the stroma: ECM anisotropy. By upregulation of LOX, CAFs increase collagen crosslinking, altering ECM topology as well as directionality and mechanical properties. Increased collagen crosslinking induces stiffer microenvironment, which modulates macrophages activation (**a**). Cancer-associated fibroblasts (CAFs) represent the main source of ECM production and remodeling within the TME where they promote neoangiogenesis and EMT (**b**). Tumor-associated macrophages (TAMs), microbiome, and extracellular vesicles (EVs) reshape ECM by secreting MMPs and matrix-associated proteins. MMPs release ECM-attached soluble growth factors and cytokines, which promote neoangiogenesis, contributing to tumor growth and potential metastatic spreading (**c**). Finally, cancer cells migrate along tension-oriented collagen fibers towards the vessels. The alignment of ECM fibers yields a rigid structure that contributes to tumor stiffness and acts as sort of highway for cancer cell migration, leading metastasis (**d**). ECM extracellular matrix, TME tumor microenvironment, CAFs cancer-associated fibroblasts, EMt epithelial–mesenchymal transition, TAMs tumor-associated macrophage, LOX lysyl oxidase enzyme, MMPs matrix metalloproteinases, EVs extracellular vesicles.

---

**Box 2 ▌ Main concepts on ECM remodeling contributing to stiffness**

Desmoplasia: increased matrix production with the remodeling of connective tissue structures adjacent to the tumor[218].
Cancer-associated fibroblasts (CAFs): the main source of ECM production and remodeling within the tumor microenvironment, promoting neoangiogenesis and epithelial to mesenchymal transformation (EMT)[19].
Lysyl oxidase (LOX): enzyme that converts lysine into high reactive aldehydes forming crosslinking between fibers of collagens and elastin[219].
Metalloproteases (MMPs): extracellular enzymes that degrade the ECM components[220].
Tumor-associated macrophages (TAMs): polarized macrophages that suppress antitumor immunity and promote tumor progression.
Anisotropy: reorganization of the topography of the ECM toward linearization of the alignment of ECM fibers.

---

non-neoplastic tissue exhibits a more random orientation of fibers and lower tissue stiffness[19,21–23]. The anisotropic organization of biological ECM is not a clinically applied parameter, but several studies have revealed that the significant increase of alignment of collagen fibers (i.e., increase of its anisotropy) has an impact on gene expression, differentiation, proliferation, and migration of cancer cells, with aligned fibers acting as sort of highway for cancer cell migration[21,26] (Fig. 2d). Interestingly, metastatic breast tumors are characterized by bundles of aligned collagen fibers oriented perpendicular to the tumor interface, highlighting the effect of ECM anisotropy as a strong regulator of directed cell migration[27,28]. Another interesting example of the effect of

tissue alignment in cancer invasion regards the pattern of invasion: invading mesenchymal melanoma and sarcoma cells, which are known to typically invade as single cells, can invade collectively when three-dimensional tissue density is high and the activity of MMPs contributes to tissue alignment[56,57]. Moreover, Friedl and coworkers demonstrated that in vivo invading cancer cells preferentially orient along aligned collagen fibers and bundles, vascular structures, and nerves[58–60], while mesenchymal tumor cells move collectively along these confined trails despite their labile cell–cell junctions[56,61]. Interestingly, anisotropy remodeling of ECM would probably lead to the generation of stiffness gradients, which leads us to introduce the concept of durotaxis. Durotaxis has recently been

proposed as a mechanism driving directed migration, which is the process by which cells follow gradients of extracellular mechanical stiffness, from soft to stiff substrate[62]. The concept of durotaxis has also been reported for several cancer cell lines, showing that cells displayed the strongest durotactic migratory response when migrating on the softest regions of stiffness gradients (2–7 kPa), with decreased responsiveness on stiff regions of gradients[63]. On the other side, a stiffer substrate promotes the proliferative capacity of cells and enhances tumor growth[64]. Durotaxis might have an important role in facilitating the forward movement of the invasion front, where tumor cells from the tumor niche move forward following the avenues of the stiffer ECM created by the surrounding CAFs.

Altogether, the increase in ECM deposition and in its alignment facilitates cell proliferation, migration, and tumor cell invasion. As a result, tumors are generally stiffer than the normal tissue, due to the ECM alterations mainly induced by CAFs[18,29]. For example, gradual stiffening of tumor stroma along with increasing tumor aggressiveness has been reported for several cancers, including colorectal and breast malignancies[65]. Another example of the association between ECM remodeling and disease is the fibrotic change in the prostate that results in an increase in the stiffness of the tissue, as a consequence of inflammatory processes[34,66,67].

Stromal cells contribute to ECM and tissue stiffness and, at the same time, a high stiffness matrix has been suggested to be important for maintaining an invasive phenotype[37], as well as favoring EMT[68] facilitating the transition from epithelial and non-motile phenotype to a mesenchymal and motile (invasive) phenotype providing cellular motility[69]. In the case of breast cancer, it has been shown that denser breast tissue is a risk factor for breast cancer development[70,71]. Such increased breast density is a result of increased connective tissue deposition and ECM components, mostly collagen[70], which is not only increased in deposition but also more oriented, stiffer, and correlated with higher epithelial cell density[71]. Collagen alignment has also been seen to have potential as a prognostic marker for invasive breast carcinoma patients[72,73]. This stiffer stroma increases breast cancer risk by inducing the oncogene *ZNF217*[71]. Furthermore, in breast cancer tumors in vivo initiation of metastasis is promoted by a stiffer microenvironment induced by increased collagen crosslinking[74]. Interestingly, it has been shown that collagen fibril density, which is a surrogate for tissue stiffness, modulates macrophage activation, and cellular functions during tissue repair[38], and that cytokine secretion is enhanced with an increase of fibril density. In a different study, it was stated that immune cells, such as macrophages are sensitive to the surrounding rigidity: lesions of higher stiffness from human breast cancer biopsies were infiltrated with a higher number of macrophages, resulting in an increased cellular TGF-b signaling[16].

**Cellular contribution to tissue stiffness: cell stiffness, density, and traction.** Not only does ECM contribute to tissue stiffness, but there is an important cellular contribution to the rigidity of their surrounding environment by means of cell stiffness, density, and contractility. As previously mentioned, cancer cells are softer than their non-transformed counterparts[75]. In vitro tests have shown that increasing ECM stiffness increases both tumor cell proliferation and invasiveness[76], although the softening of cancer cell lines was not associated with the aggressiveness of several solid neoplasia[75,77–79]. Of particular note is the fact that the elasticity of tumor cells now represents one of the markers for the recognition and isolation of circulating tumor cells[80].

Although it cannot be excluded that a stiffer substrate induces a softening of the tumor cells, it also cannot be excluded that a

softening of the tumor cell is an early event associated with the malignant transformation. Indeed, it has been reported that mechanical forces played by the neoplastic cells or cancer-associated fibroblasts in the in situ carcinoma are sufficient to remodel the basal membrane to allow for tissue invasion[81]. On the other hand, it has been reported in vitro that focal adhesion points of the cell with the substrate, resulting in the formation of stress fibers in the cell, mediate cell traction. When cell density is high enough to keep cells close to each other, this cell traction increases the local stiffness, an event that is particularly relevant when high density of tumor fibroblasts or mesenchymal stem cells are seeded on a soft substrate[82].

Tumor progression is mediated by the ability of tumor cells to invade the tissue layers to reach the vessel and spread in secondary organs. Invasion by in situ carcinoma requires tumor cells to pass the basal membrane to reach the lamina propria located below the epithelial layer. Besides the MMP-mediated remodeling of the ECM, including the ECM composing the basal membrane[83], physical remodeling of the basal membrane has also been reported to be induced independently of MMPs. The basal membrane can be remodeled by a large protrusion of cancer cells that physically tears and then displaces the basal membrane[84]. On the other hand, CAFs have recently been reported to modify the ECM of the basal membrane independently of MMPs[81]. In the presence of MMP inhibitors, the physical contact between CAFs and the basal membrane was reported to sustain invasion. By applying mechanical forces on the basal membrane, CAFs reduced the density of fibers of intermediate stiffness, creating patches of soft, inhomogeneous material sparsely interspersed with thick fibers. By exerting contractile forces CAFs can soften the basal membrane, thus adding a second mechanism that is proteolysis-independent to the tumor progression of in situ carcinoma. Because this event has not been reported for fibroblasts isolated from the juxta-tumoral tissue, it would be important to unveil the tumor-derived factors (i.e., extracellular vesicles (EVs)) that induce transformation of fibroblast to CAFs and that enhance their mechanical forces.

Increased local stiffness by mechanical forces due to cell traction, proteolysis-independent remodeling of the ECM adds another piece of information regarding tumor progression and why MMP inhibitors have failed in clinical trials.

**The pre-metastatic niche.** Tumor cells, together with CAFs, hematopoietic progenitor cells, and TAMs prepare a suitable "soil" for incoming metastasis in distant tissues and organs. This abnormal, tumor growth-favoring microenvironment is the so-called pre-metastatic niche[85]. The ECM is closely related to tumor metastasis, and the tumor-environmental transition from softer tissue to stiff fibrous tissue goes hand in hand with metastasis progression. First, in the primary tumor site, primary tumor cells secrete soluble growth factors (as VEGF-a, TGF-b and tumor necrosis factor alpha) and EVs (see section 4.1) containing miR-NAs, integrins, growth factor receptors, and chemoattractants[86,87]. These soluble growth factors and EVs arrive at the target tissue and prime it, as a first step of preparing a suitable micro-environment for metastasis formation[88]. As a second step, bone marrow derived cells, CAFs, myofibroblasts and TAMs colonize the niche, resulting in an altered expression of collagens and MMPs. Finally, tumor cells that underwent EMT arrive at the target tissue as tumor circulating cells, which they colonize to form a second tumor[14,89,90]. Collagen stabilizing enzymes are over-expressed during metastasis and hypoxic conditions, where they play a critical role hydroxylating collagen in response to hypoxia-inducible factors[14]. One example of the priming of the tissue is

breast cancer cells that metastasize to lung, where they secrete collagen and change the organization of ECM fibers[13,19]. Color-ectal liver metastasis presents high collagen turnover together with collagen isoforms changes, contributing to the generation of a favorable tumor microenvironment[91,92]. On the other hand, metastatic melanoma cells can re-organize the collagen matrix depending on their invasive potential: the higher the invasive potential, the greater the traction force of the cell on the ECM, resulting in a more linearized ECM where cell migration is enhanced[93].

### Novel mediators of ECM remodeling

**The extracellular vesicles**. EVs are small membranous vesicles released into the extracellular environment by virtually all cell types and they act as a "cell-to-cell delivery service" since they contain cell-specific cargo molecules, including proteins, lipids, and nucleic acids[94]. EVs secreted by both cancer cells and cancer-associated cells, and in particular exosomes (i.e. a particular group of EVs characterized by specific biogenesis and diameter), were shown to transfer several bioactive molecules to other recipient cells, inducing modifications of their environment and facilitating tumor growth, invasion, metastasis and the formation of the pre-metastatic niches[95–98].

A recent finding by Huleihel showed that biologic scaffold materials contain active EVs bound by matrix components, which strengthen the general role of EVs as structural and functional components of the ECM[99]. Huleihel reported that EVs are closely associated with the collagen network in biological scaffolds. Moreover, EVs are able to exert autocrine control of directional cell migration in vivo by promoting cell polarization and the assembly of adhesions with the ECM components, such as collagen fibers, via integrin–fibronectin interactions[100]. Similarly, Villasante and coworkers showed that the microenvironmental signal, including the three-dimensionality, composition, and stiffness of the tumor matrix, are all together necessary for recapitulating the properties of exosomes found in the primary tumor[101].

EVs also contain proteinases, such as MMPs, which assist cell invasion during angiogenesis and cancer progression[102]. Tumor-derived EVs deliver ECM-MMPs inducers which can contribute to matrix degradation in different diseases[103,104]: For instance, EVs derived from ovarian cancer contain high levels of MMP-2, MMP-9, and urokinase-type plasminogen activator proteinases that could degrade the ECM[104]. Similarly, MMP-1, MMP-3, and MMP-13 were reported to promote angiogenesis, tumor pro-liferation, and invasion also in prostate cancer (PCa)[105]. Likewise, fibroblasts migrating into the collagen matrix were documented to release EVs that carry MMP-9 to act upon the collagen fibrils, which leads to a gradual transformation of the collagen matrix from a laminar to a fibrillar type of architecture[91]. All these phenomena might be, from one side, responsible for the pro-invasive and the pro-angiogenic activities mediated by cancer cells or, on the other side, could also strengthen the interaction of EVs with the matrix itself.

A key step during cancer progression to metastasis is achieving the ability to migrate and invade. It has been demonstrated in bladder cancer that differences in mechanical and complementary activation properties of malignant and non-malignant cell-derived EVs may contribute to increase endothelial disruption as well as tumor growth[92]. Furthermore, endothelial leakiness induced by malignant EVs might be a pre-condition for cancer cell trans-endothelial migration during the metastatic progression[92].

As tumors progress, they need to recruit new blood vessels to ensure a sufficient supply of oxygen and nutrients; this process requires the release of neovascularization-stimulating factors[93]

and EVs have been implicated in this process. Hypoxia, which impact angiogenesis, tumor progression, and immune tolerance, has been shown to increase EVs release from cancer cells[106,107]. Secondly, the presence of matrix MMPs also allows for the direct modulation of the ECM by EVs in the primary tumor microenvironment and metastatic spread, contributing to ECM degradation to provide spaces for blood vessel recruitment. In renal cell carcinoma, EVs were enriched with azurocidin protein, which is involved in vascular permeabilization[108]. Furthermore, EVs released by bladder-cancer patients were shown to be enriched with EDIL-3, which activated epidermal growth factor receptor signaling in cancer and endothelial cells, promoting their angiogenesis and migration[95]. Similarly, PCa cell-derived EVs contain TGFβ and stimulate the differentiation of bone marrow mesenchymal stem cells into myofibroblasts, which secrete high levels of VEGF-A, HGF, and MMPs, such as MMP-1, -3, and -13, have pro-angiogenic functions and enhanced tumor cell pro-liferation and progression[105].

As reported above, another key event during cancer progres-sion is the formation of the pre-metastatic niche. Recently, several studies have reported that the formation of a pre-metastatic niche depends on tumor-derived EVs[109–111]. EVs increase angiogenesis and vascular permeability in the pre-metastatic niche[112–114] and determine organotropic homing[90]. For instance, exosomes derived from high-grade bladder cancer educate pre-metastatic niches and facilitate distant metastasis[89]. Similarly, human kidney cancer stem cell-derived CD105-positive microvesicles can also stimulate angiogenesis, thereby promoting the formation of the pre-metastatic niche[112].

**The microbiota**. Among the potential modulators of the ECM properties, the microbiota—i.e., the collection of all micro-organisms (bacteria, viruses, and fungi) found in a specific environment—certainly plays a role[115]. Every tissue in the human body has a defined ecological community of bacteria associated with good health (eubiosis), whose disruption—a condition known as dysbiosis—has been central in a variety of diseases, including periodontal pathologies, inflammatory diseases, and cancer. It is worth mentioning that we never refer to individual bacterial communities when talking about the resident human microbiota, but rather to a complex ecosystem (i.e. a biofilm) in which different bacterial entities are embedded in a complex ECM and interact with the host[116]. Bacteria produce a variety of proteases that control biofilm homeostasis and, at the same time, may interact with the host counterpart, mediating host ECM degradation, collagen crosslinking, and cell biomechanical properties[117]. It can be hypothesized that these bacterial enzymes have a role in the physiological and pathological turnover of the host tissue ECM, depending on the status of physiological eubiosis vs. pathological dysbiosis of the microbiota itself. Among these enzymes, elastases, hyaluronidases, alkaline phosphatases, and phospholipase C have been thoroughly studied, especially in the context of infection[118–120]. For instance, bacterial elastases have been shown to degrade the ECM and, by interfering with the turnover of interferon-γ, interact with lymphocyte proliferation and activation[117,121].

Another class of important ECM-degrading enzymes is represented by bacterial collagenases, which are able to digest the triple helix region of collagens with a broader substrate specificity compared to the eukaryotic counterparts, promoting bacterial spread and host tissue modifications[122–126]. Bacterial collagenases are responsible for tissue degradation, an important step for colonization, invasion, and dissemination in the host tissue, and for the acquisition of nutrients for growth and proliferation. Interestingly, abnormal collagen turnover has been

described as a pathological hallmark in many human conditions, such as cancer, arthritis, and atherosclerosis, in which the resident bacterial flora plays a physiopathological role[127,128]. Likewise, bacterial dysbiosis might also contribute to impaired tissue homeostasis and thus onset/progression of fibrosis. Significant reduction of *Clostridia* was reported in human testis with idiopathic germ cell aplasia, which was characterized by fibrosis of the seminiferous tubules[129,130]. Noteworthy is the fact that *Clostridia* releases collagenases that only digest interstitial collagens type I, II, and III[131]. One possible interpretation of this association is that continued collagen production by eukaryotic cells is not counterbalanced by the degradation by collagenases from tissue-resident clostridia, helping to exacerbate the progression of fibrosis.

As described above, LOXs are another class of enzymes important for the mechanobiology of the ECM[132]. In particular, LOX-coding genes have been found in the genome of five major bacterial clades: *Bacteroidetes*, *Actinobacteria*, *Proteobacteria*, *Gemmatimonadetes*, and *Deinococcus-Thermus*[133,134]. It can be hypothesized that bacterial LOX may contribute to the modifications of the matrix stiffness in malignancies in which the resident microbiota has been involved, thus collaborating with CAFs and tumor cells to establish a microenvironment beneficent for cancer growth and dissemination[34,66,135–137].

Bacterial enzymes are also implicated in post-translational modifications of host proteins[138]. For instance, *Porphyromonas gingivalis* expresses a peptidyl-arginine deaminase able to citrullinate collagen I, modifying the interaction between collagen I and fibroblasts expressing integrin $\alpha_{11}\beta_1$, thus contributing to the development and progression of destructive arthritis[139,140]. Despite being plausible to hypothesize a similar role of bacteria-induced post-translational modifications of ECM proteins in the onset and progression of solid malignancies, this has not yet been demonstrated, opening the possibility to further investigations.

So far, we have analyzed how bacteria may contribute to modify the ECM microenvironment to favor the onset or progression of pathological conditions. However, it has been recently reported that the host ECM is also able to influence bacterial behavior and function, in particular the bacterial ability to invade and infect mammalian host cells. By manufacturing hydrogels of varying stiffness seeded with human microvascular endothelial cells, Bastounis et al. found that adhesion of the bacterial pathogen *L. monocytogenes* to host cells increases monotonically with increasing matrix stiffness[141].

Based on these findings, it seems feasible that the interplay between tissue-resident microbiota and host ECM has a bidirectional relationship of reciprocal influence[135]. During eubiosis, bacterial enzymes may play a role in the physiological turnover of the host ECM, which in turn conveys certain growth signals back to bacteria that maintain a "healthy" and noninvasive bacterial ecosystem. Conversely, in case of dysbiosis, the enzymes produced by the dysbiotic bacteria may contribute to polarize the ECM towards a "malignant" phenotype able to promote the onset and progression of host pathological conditions. At the same time, the "malignant" ECM may contribute to increase the ability of bacteria to invade the host tissue, infect host cells and produce genotoxins and other virulence factors. Future studies are needed to establish the relevance of this hypothesis.

## Organ-specific stiffness

**From micro to macroscale: stiffness in clinical practice**. In the next sections, we will review the current knowledge about stiffness in benign and malignant urological diseases. Before proceeding forward, it should be clear to the reader that the different available techniques to study tissue and organ stiffness vary in the

resolution of the appreciable modifications (Table 1). Atomic force microscopy (AFM)-based measurements provide the finest resolution in the micro-/nano- scale, but they are limited to ex vivo specimens and by a long acquisition time, making its translation into clinical and diagnostic practice difficult. On the other hand, imaging-based techniques, like magnetic resonance elastography (MRE), shear wave elastography (SWE), and real-time elastography, can only provide macroscale maps of tissue stiffness, thus providing an overview of the organ stiffness. How to complement different techniques with different resolutions is still a matter of research, and future studies are needed to define the framework to integrate micro- and macro-scale stiffness data.

**Tissue stiffness—kidney**. Histology of the kidney affects its mechanical properties, particularly the amount of fibrosis in the parenchyma[142]. Pathological findings of glomerulosclerosis and tubulointerstitial fibrosis obtained through biopsies are associated with poor prognosis in renal diseases, and with lower renal parenchymal elasticity[143]. Renal biopsy, although being the gold standard for assessing fibrosis with histological techniques, is an invasive procedure that can cause complications. In order to develop early diagnostic tools, there is huge interest in developing noninvasive methods to accurately evaluate nephropathy.

SWE provides quantitative information of the average tissue elasticity of the whole organ, based on the assumption that the structure of benign tissue differs from that of cancerous tissue. Therefore, SWE can be used to compare the stiffness of benign and malignant tissue, and establish a threshold of elasticity, i.e., of Young's modulus, from which the tissue is considered malignant. In patients with chronic kidney disease the renal parenchymal stiffness was measured by SWE, and the obtained Young's modulus was correlated with serum creatinine, urea levels, and the estimated glomerular filtration rate (eGFR). A positive correlation between stiffness and age, serum creatinine, and urea was observed, while the eGFR level had the opposite trend and was negatively correlated with stiffness[142]. The same study also established that a value of tissue stiffness higher than 4.31 kPa corresponded to a diseased kidney (Table 2), an information that can be used to develop early noninvasive diagnostic tools. Of interest is also the fact that a correlation has been reported between renal stiffness and degree of renal fibrosis[144,145], with patients suffering from chronic kidney disease exhibiting a stiffer renal parenchymal than non-chronic kidney disease patients[146]. Renal elasticity is therefore a potential predictor of chronic kidney disease[143]. In fact, patients with later stages of chronic kidney disease had stiffer renal cortex, with stiffness increasing progressively from stage 3 to 5 of the disease. Interestingly, a worsening of renal stiffness was associated with proteinuria, which is characterized by the infiltration of inflammatory cells into the renal interstitium and the replacement of the tubulointerstitium with fibrous scar[147]. The worsening of renal elasticity was associated with rapid renal deterioration in these patients, suggesting that proteinuria could be an indicator of renal elasticity and early renal fibrosis. Similar results were observed when comparing renal stiffness values obtained by MRE, which differentiated patients with CKD (5.10 kPa) from those with normally functioning kidneys (4.35 kPa). The mean stiffness in patients with CKD significantly increased from stage 1 to stage 4[148]. At stage 5 CKD (kidney failure) the renal stiffness decreased, likely due to renal hypoperfusion that can mask fibrosis[148,149]. Furthermore, MRE was also evaluated for assessing the chronic renal allograft dysfunction. The skewness of MRE corticomedullary stiffness was sensitive to changes in chronic allograft dysfunction, and the MRE cortical and corticomedullary mean stiffness appears to be a predictor of graft loss/relist; a cutoff of

**Table 1 Commonly used techniques for the mechanical characterization of living tissues.**

| Technique | Concept | Modulus | Sample | Scale | Ref |
|---|---|---|---|---|---|
| Atomic force microscopy (AFM) | Atomic-level indentation (nanoindentation) or shear rheology (atomic force microscopy-based rheology) | E (indentation), G', G" (shear) | Ex vivo tissue | Microscale, nanoscale | 209–211 |
| Shear rheometry | Application of small-amplitude oscillatory shear stress and quantification of the resulting strain | G', G" (shear, viscoelastic) | Ex vivo tissue | Macroscale | 212,213 |
| Compressive deformation | Classic stress–strain analysis. Uniaxial stress is applied to compress the material and a relationship is established with the resulting strain | E (elastic) | Ex vivo tissue | Macroscale | 214 |
| Magnetic resonance elastography (MRE) | Magnetic resonance visualization of tissue deformation resulting from the introduction of shear waves into the tissue derived from external vibrations noninvasive, promising for clinical applications | G', G" (shear, viscoelastic) | In vivo tissue, | Macroscale | 215,216 |
| Real time elastography (RTE) | Sonography-based noninvasive method. It uses conventional ultrasound probes to compare echo signals before and after slight compression | E (elastic) | In vivo tissue | Macroscale | 217 |
| Shear wave elastography (SWE) | External acoustic force pulses are used to generate shear waves which propagate perpendicular to the ultrasound beam, causing transient displacements that result in an image of the distribution of the shear-wave velocities | Shear wave speed (SWS), that can be converted into E and G | In vivo tissue | Macroscale | 218 |

Adapted from Guimarães et al. [8].

2.48 and 3.29 kPa for no progressive decline was identified on cortical and corticomedullary stiffness, respectively[150]. In a preclinical model of atherosclerotic renal artery stenosis, MRE was also used to monitor decreased medullary stiffness in response to low-energy shockwave therapy[151].

Besides MRE, quantitative magnetic resonance imaging (MRI) using tomoelastography has been used to assess renal parenchyma softening in IgA nephropaty (shear wave speed 1.86 m/s vs. 2.34 m/s in age-matched healthy volunteers)[152]. The same authors also stated that a cutoff value of 2.05 m/s was used with a sensitivity and specificity of 81% and 100%, respectively, and identified a positive correlation of shear wave speed and eGFR, still supporting that renal parenchyma softening occurs when kidney perfusion is decreased[152].

Furthermore, renal elasticity is also a potential predictor for renal transplant outcome[153–157], as well as an indicator for diabetic nephropathy[146]. Although tissue biopsy is the gold standard for assessing renal fibrosis, this procedure is invasive and prone to sampling error. Imaging techniques, such as MRE, MRI, and SWE constitute promising noninvasive techniques to monitor mechanical changes associated with renal function.

**Kidney specific mechanisms for increased stiffness**. Kidney stiffness is associated with kidney fibrosis, age, proteinuria, and poor prognosis. Fibroblasts play a major role in kidney fibrosis, and an important accumulation of fibroblast was reported in the kidney, with 15% of fibroblasts in renal fibrosis originating from bone marrow, 36% from local tubular epithelial cells via EMT and the remaining 50% from proliferating resident fibroblasts[158,159]. Furthermore, Activin A has been reported as a relevant stimuli which induced the expression of fibronectin and collagen I in renal interstitial fibroblasts[19,160]. EMT is crucial for organ fibrosis, i.e., deposition of collagens, elastin, tenascin, and additional matrix proteins. Fibroblast-specific protein 1 (FSP1) is a very interesting protein found in the EMT proteome, whose expression correlates with EMT during kidney fibrosis[161]. Inhibition of FSP1 attenuates fibrosis and collagen deposition, and preliminary studies suggest that FSP1 may sequester p53 from the APC ubiquitination pathway, resulting in an increase of β-catenin levels, and thus, facilitating EMT phenotype.

**Tissue stiffness—bladder**. The mechanical properties of the bladder are essential for its function: it has to adapt and stretch to the urine volume it contains and undergo high elastic extension. Although alterations of the mechanical properties of the bladder result in a dysfunction of its physiological role, the mechanical properties of bladder tumors have been scarcely studied. Nevertheless, most benign bladder pathologies are associated with an increase of ECM-fibrosis and may progress from the formation of stiffer matrix to a more compliant structure[162]. Such mechanical information is very important and relevant for several purposes, as is the development of innovative diagnostic tools, computational models, improvement of surgical devices, and surgical trainers.

Already in 1994, bladder wall elasticity was classified as a physiological biomechanical characteristic, susceptible to change with the development of different pathologies[163]. In this study, they measured the ratio of connective tissue to smooth muscle in patients with bladder pathology and compared them to normal bladders. They found that such ratios were increased compared to normal tissues, indicating that in the dysfunctional bladder there is a higher proportion of connective tissue compared to smooth muscle tissue. Although no mechanical tests were performed, they directly correlated these observations with a loss of elasticity in the bladder wall. Interestingly, later published studies have

**Table 2 Mechanical moduli of human urological tissues.**

| Tissue | Technique | Modulus | Modulus value (condition) | Ref |
|---|---|---|---|---|
| Kidney | | | | |
| | SWE | YM | 4.31 kPa (healthy) | 142 |
| | RTE | | 75.1 ± 37.8 (non-CKD) | 143 |
| | | | 72.9 ± 37.6 (CKD stage 3a) | |
| | | | 59.3 ± 40.3 (CKD stage 3b) | |
| | | | 48.3 ± 33.8 (CKD stage 4) | |
| | | | 36.6 ± 33.0 (CKD stage 5) | |
| | MRE | YM | 4.35 kPa (normal functioning kidneys) | 148 |
| | | | 4.86 kPa (cutoff value) | |
| | | | 5.10 kPa (CKD patients) | |
| | MRE | YM -Cortico-medullary | 3.24 kPa (functional allografts) | 150 |
| | | YM -Cortical | 3.73 kPa (chronic dysfunction allografts) | |
| | | | 3.29 kPa (no progressive decline) | |
| | | | 4.82 kPa (graft loss/relist) | |
| | | | 2.43 kPa (functional allografts) | |
| | | | 2.84 kPa (chronic dysfunction allografts) | |
| | | | 2.48 kPa (no progressive decline) | |
| | | | 3.67 kPa (graft loss/relist) | |
| | MRE | SWS | 1.86 m/s (IgA neprophaty) | 152 |
| | | | 2.05 m/s (cutoff value) | |
| | | | 2.34 m/s (healthy) | |
| Prostate | | | | |
| | SWE | YM | 42 kPa (cutoff value healthy) | 177,180 |
| | SWE | YM | 144.85 kPa (cutoff value for recurrence after radical prostatectomy) | 182 |
| | SWE | YM | 31.79 ± 16.17 kPa (benign) | 177 |
| | | | 114.96 ± 85.25 kPa (malignant) | |
| | SWE | YM | 95 ± 28.5 kPa (Gleason score 6) | 219 |
| | | | 163 ± 63 kPa (Gleason score 7) | |
| | SWE | YM | 91.6 kPa (Gleason score 6) | 220 |
| | | | 102.3 kPa (Gleason score 7) | |
| | | | 131.8 kPa (Gleason score 8) | |
| | SWE | YM | 32.7 ± 19.4 kPa (Gleason score 6) | 179 |
| | | | 55.4 ± 48.5 kPa (Gleason score 7) | |
| | | | 57.3 ± 39.4 kPa (Gleason score 8) | |
| | | | 88.2 ± 64.2 kPa (Gleason score 9) | |
| | MRI with tomoelastograhy | SWS | 2.8 ± 0.4 m/s (peripheral zone) | 185 |
| | | | 2.8 ± 0.3 m/s (transition zone) | |
| | | | 3.1 ± 0.6 m/s (PCa) | |
| Bladder | | | | |
| | Rheology | Storage modulus | 0.052–0.085 Mpa (healthy) | 165 |
| | | Loss modulus | 0.019–0.043 Mpa (healthy) | |
| Testis | | | | |
| | SWE | Velocity stiffness | 0.76 m/s (normal) | 188 |
| | | | 0.79 m/s (testicular microlithiasis) | |
| | | | 1.92 m/s (testicular cancer) | |
| | SWE | Velocity stiffness | 0.77 m/s (normal) | 189 |
| | MRI | Diffusion values | 0.78 m/s (testicular microlithiasis) | |
| | | | 1.95 m/s (testicular cancer) | |
| | | | $0.929 \times 10^{-3}$ mm$^2$ s$^{-1}$ (normal) | |
| | | | $0.978 \times 10^{-3}$ mm$^2$ s$^{-1}$ (testicular microlithiasis) | |
| | | | $0.743 \times 10^{-3}$ mm$^2$ s$^{-1}$ (testicular cancer) | |
| | SWE | YM | 4.5 kPa (cutoff value for semen parameters improvement after surgery) | 190 |
| | SWVV | | 1.465 m/s (cutoff normal/oligozoospermia) | 192 |
| | | | 1.328 m/s (cutoff normal/azoospermia) | |
| | SWE | YM | 2.50 ± 0.49 kPa (Varicocele grade I) | 191 |
| | | | 2.59 ± 0.81 kPa (Varicocele grade II) | |
| | | | 2.80 ± 0.72 kPa (Varicocele grade III) | |

*CKD* chronic kidney disease, *RTE* real-time elastography (uses arbitrary units), *YM* Young's modulus, *SWS* shear-wave speed, *SWVV* shear wave velocity values, *MRI* magnetic resonance imaging diffusion values.

suggested to use the ratio of fibrous connective tissue to smooth muscle tissue measured by SWE as a parameter for the diagnosis of early fibrotic changes[164].

The viscoelastic properties of human bladder tumors at the macroscale have been tested by different techniques. In one study[165], researchers analyzed ten bladder human tumors by dynamic mechanical analysis. Samples were collected from patients by transurethral resection procedures, and they applied a load to the tumor with increasing frequencies, up to 30 Hz, calculating both the storage and loss modulus (Table 2). These macroscopic measurements were previously established on porcine bladder by the same group[166]. This analysis quantifies the frequency-dependent viscoelastic properties on the macro-scale which is rheometer, getting two different mechanical parameters for each tumor: storage modulus, ranging between 0.052 and 0.085 MPa; and loss modulus ranging between 0.019 and 0.043 MPa.

Efforts are being made to develop new techniques to monitor changes in bladder wall mechanical properties in a noninvasive way[167]. Aiming to develop noninvasive diagnostic tests for lower urinary tract disorders (LUTS), one study compared three quantifiable ultrasound methods (high-frequency ultrasound, SWE, and duplex doppler) to measure the biomechanics of the bladder wall in healthy individuals, in order to establish baselines and reference points for future research. Such quantitative noninvasive diagnostic tool would allow to detect bladder wall changes and decreased wall function before obvious fibrotic changes develop[164]. They observed that bladder wall pathology

affects the structure and thickness of the bladder wall layers; and the thickness of the bladder and detrusor layer increased with age. The increase in thickness with age could be related to an increased interstitial collagen deposition or to hypertrophy of the detrusor.

Mechanical properties of bladder cancer have been studied at different scales. At a cellular level, also mechanical properties are of relevance, and there is increasing evidence that the mechanics of cells can be used as a marker for pathology (metastatic potential, differentiation degree, etc.). Bladder cancer cells with different metastatic potential have been characterized[168], showing that cell lines with higher aggressiveness have a lower Young's modulus than lower grade cancer cells. These observations have been reported by several studies[75,169–172], showing that cancer cells are more elastic than their benign counterparts. This increase in elasticity (indicated by a decrease in the cell's Young's modulus) means higher deformability of cancer cells, which could have implications for facilitating intra and extravasation and thus, metastasis. This tendency has been observed not only in bladder cancer cells, but also in tissues, such as prostate and breast.

**Bladder specific mechanisms for increased stiffness.** Changes in the bladder wall associated with a loss of bladder elasticity and dysfunction could be a consequence of inflammation, loss of urothelium and obstructive or neurogenic etiologies. These processes would lead to hypertrophy of smooth muscle cells, hyperplasia of fibroblasts and the deposition of collagen fibers between muscle bundles of detrusor, and bladder wall thickening, and consequently result in fibrosis, scarring and stiffening of the bladder, together with a progressive reduction in bladder capacity[164,173].

The ECM plays a fundamental role in cancer progression and metastasis. In particular, collagen stiffness has been proposed to promote BCa progression from non-muscle invasive bladder cancer (NMIBC) to muscle invasive bladder cancer (MIBC), specifically six collagen family members located in the ECM-receptor interaction signaling pathway[66,174].

Furthermore, the expression of these six collagen genes is negatively correlated with a promising prognosis of BCa patients, overall survival rates, and recurrence-free survival. Therefore, these genes could be considered high progression risk factors and be used as independent effective diagnostic and prognostic biomarkers for BCa, suggesting their potential as targets for clinical treatment.

Interestingly, the linearization and topography of stroma have been seen to differ in MIBC patients[13]. Neoplastic ECM was observed to be more dense and compact, and had an increase of linearization of fibers, as similarly observed in on colorectal[175] and ovarian cancer[176]. Furthermore, there was a loss of tissue morphology and increased vascularization.

**Tissue stiffness—prostate.** PCa has a higher cell and vessel density, which is discernibly stiffer than both benign and normal tissues[177]. Given the noninvasive and cost-effective imaging technology of SWE, several studies have been published aiming to evaluate differences in stiffness between PCa and benign tissue based on Young's modulus[178] (Table 2). For all studies discussed here, it was reported that the SWE-obtained Young's modulus of PCa was significantly higher for benign prostate tissue. The stiffness of PCa in the peripheral area, where most PCa starts, gradually increases with the Gleason Score, which is the grade of the tumor: the higher the Gleason score, the stiffer the tissue is[178,179]. The stiffness values corresponding to each Gleason score differ when reported by different studies. For example, the Young's modulus value to use as a threshold to discriminate

benign from malignant prostate tissue has been reported to be 42[180], 35[181], or 31 kPa[177]; and a value of 144 kPa has been proposed as a predictor of biochemical recurrence following radical prostatectomy[182].

It is important to keep in mind that the prostate is a mechanical heterogeneous tissue: its overall elasticity can vary greatly from zone to zone. In addition, there is patient to patient variability in stiffness. Nevertheless, given the fact that SWE allows for the calculation of elasticity ratios between benign and malignant tissue, this technique provides a standardization and results in user-independent imaging of the prostate. Despite inter-variation among the studies, they all agreed on the utility of measuring prostate stiffness by SWE, as this provides additional information for the detection and biopsy guidance of PCa, enabling a substantial reduction in the number of biopsies while ensuring that few peripheral zone adenocarcinomas are missed. SWE can provide information on tissue elasticity, but the combination of this technique with magnetic resonance can improve the detection of PCa[183] and provide guidance for more targeted biopsies rather than systematic ones, thus increasing the positive rate of PCa in targeted biopsies[177,184]. The biophysical signature of PCa, characterized by increased stiffness, reduced water diffusion, and increased mechanical fluidity of cancer tissue, correlates with increased cell density and fibrous protein accumulation[185]. SWE as well as tomoelastography with diffusion-weighted MRI have great potential for diagnosis, as they can provide quantitative maps of tissue mechanics. With these data, we can conclude that prostate stiffness (obtained by SWE and/or MRI) can be used as a significant marker to enhance the predictive ability of other clinical histopathological factors for PCa detection and diagnosis in clinical practice.

Prostate stiffness seems to play a role in the development of urological syndromes, such as LUTS[67]. LUTS is indicative of benign prostatic hyperplasia and prostatic calculi are closely associated with reduced urinary flow rate and LUTS severity. In general, most calculi are associated with inflammation of the prostate, characterized by lymphocyte and histiocyte infiltration that will feed the inflammatory loop. This chronic inflammation, through the development of fibrosis and calcification will affect urethral stiffness. Prostatic fibrosis and urethral stiffness have been suggested as potential etiological factors of LUTS[186]. In an additional study, periurethral tissue stiffness from radical prostatectomies were mechanically tested by uniaxial load-unload, showing that the periurethral tissues from patients with LUTS were significantly stiffer and had higher collagen content compared to the periurethral tissue of patients without LUTS[181]. Another group suggested that prostate inflammation can induce fibrotic changes in periurethral prostatic tissues, resulting in increased urethral stiffness and LUTS[187]. Diabetes also seems to induce inflammatory changes that can be associated with the development of prostatic fibrosis[186].

**Prostate-specific mechanisms for increased stiffness.** High periurethral stiffness has been correlated with worse urinary symptoms[67]. An explanation of the correlation between the stiffness of the periurethral tissue and LUTS is that an increase in tissue stiffness can be a consequence of a fibrotic change, reducing elasticity and functionality of the tissue. When fibrosis occurs, tissue stiffness results from myofibroblast accumulation, collagen deposition, and ECM remodeling. It has been suggested that calcification and fibrotic changes in the urethra are a consequence of inflammatory processes around the prostatic urethra[186]. Altogether, these results suggest that prostatic fibrosis increases urethral stiffness, resulting in decreased urethral flexibility that compromises the function of prostatic urethra during micturition[187].

**Tissue stiffness—testis**. Modulation of the tissue stiffness has also been reported in several pathologies of the testis. The stiffness of testicular cancer can be more than double the stiffness of normal testis, suggesting that increased stiffness could be used as a testicular malignancy marker, and detectable by ultrasound elastography[188,189]. In testicular microlithiasis, a modest increase of tissue stiffness compared to normal testicles was measured by SWE[188], but no malignant characteristics are measured by SWE or MRI diffusion in this condition (stiffness values reported in Table 2)[189]. Thus, benign testicular lesions can be differentiated from malignant ones both by MRI diffusion and elastography. Testis stiffness has also been investigated in the varicocele, where collagen deposition and interstitial testicular edema have been described[190]. Of interest was the predictive potential of SWE for patients undergoing varicocelectomy, which showed a negative correlation of testis stiffness after surgical intervention and improvement of semen analysis parameters (sperm count and motility)[190–192], pointing to the potential of using SWE tissue stiffness measurements to predict fertility improvement.

**Testis specific mechanisms for increased stiffness**. A possible mechanism for the development of testicular fibrosis and increased tissue stiffness has been described. Inflammation and immunological factors contribute to testicular damage: infertile patients exhibit interstitial immune cell infiltration, loss of germinal epithelium, a thickening of the lamina propria and fibrosis of the seminiferous tubules[193]. Activin A seems to play a fundamental role in regulating inflammatory responses of the testis and the development of testicular fibrosis[159] and it is widely expressed in the testis under physiological conditions, where it is mainly produced by Sertoli cells. Activin A regulates fibrosis by stimulating fibroblast proliferation and differentiation into myofibroblasts. It also activates the transcription factor SMAD2 by phosphorylating it, promoting the transcription of several ECM genes. *Activin A* expression is increased in human testes with leukocytic infiltrate and impaired spermatogenesis, and its expression correlates with severity of the disease. Furthermore, these patients present increased collagen and fibronectin deposition, and thickening of basement membrane and lamina propria. It is hypothesized that fibroblasts and peritubular cells contribute to testicular fibrosis after inflammation, and activin A is implicated in such a process. Therefore, activin antagonists might have beneficial effects in limiting inflammation and fibrosis to treat testicular disease, as reported for kidney[160].

**Reconciling the contribution of cellular and microenvironment modifications with macroscale observations**

The mechanomedicine research field is a rapidly growing area of medicine, although there are a few limitations that will need to be addressed in the near future:

(1) Due to the diversity in the techniques and methodologies of the protocols and procedures used by different groups, there is no standardized methodology to assess tissue and organ mechanics[7,8,194].

(2) Working with non-fixed living cells and tissues in the clinical environment continues to be a challenge for the study of cell and tissue mechanics. When a biopsy is performed, for example, in the bladder through cystoscopy, the amount of tissue is very limited and in most cases the amount collected is required entirely by the pathologists to perform their diagnosis. Furthermore, the standard procedure of fixation that tissues undergo after being biopsied is necessary to maintain all histological information as

accurately as possible[195], but creates difficulties for mechanical tests. It is therefore crucial that noninvasive methodologies, such as SWE, are utilized. Furthermore, there are studies for the detection of bladder cancer measuring the mechanical properties of cells collected from urine[196].

(3) On the other hand, we are aware of the gap between proposed molecular mechanisms and reported stiffness values. We have aimed to describe the molecular mechanisms on a microscale: the increase of ECM deposition, the remodeling and reorganization of its components, anisotropy, the effects of cell traction, contractility, proposed EVs, and the microbiome as novel players regulating the molecular processes that ultimately determine tissue stiffness at the organ scale, the macroscale. While macroscale measurements provide an overview of structural modifications, the underlying events/mediators cannot be elucidated through techniques such as elastrography. AFM, optical and/or magnetic tweezers, as well as microaspiration, each of which provides mechanical information at the microscale, may be useful in unveiling the contributions of several variables leading to the modification of the tissue properties. For this purpose, the development and/or implementation of new innovative in vivo microscale mechanical techniques is of critical importance. Linking the observed macroscale stiffness to microscale contributors might potentially drive the development of early diagnostic tools and, more interestingly, prognostic tools that could allow for the identification of regions at risk for relapse of malignancies, i.e., bladder relapse which occurs in up to 70% of NMIBC[197].

**Conclusions**

Aging, neoplasia, and metabolic diseases can lead to fibrosis and an increase of tissue stiffness[5]. Thus, tissue stiffness is a clinically relevant parameter for diagnosis and prognosis purposes, as we have here shown that alteration of this mechanical parameter is coupled with the progression of diseases and malignancies associated with kidney cancer and fibrosis[143], PCa[177,178], bladder cancer[167], testicular cancer[189], and infertility related diseases[190–192]. Biochemical, topographical, and mechanical modification of the ECM occurs both at the primary and metastatic site[19,21,26], but there is also an important cellular contribution to the altered tissue stiffness by means of cell density, contractility, and traction[82,93]. At the same time, there is a crosstalk between ECM and cells, being this regulation not only biochemical but also physical, as it has been seen that a stiffer ECM induces cell proliferation and EMT, and favors the migration of the invasion front of the tumor[68]. There are several factors that regulate and remodel ECM, with CAFs being one of the most important players. Furthermore, there is a MMP dependent and independent modification of the ECM that contributes to tumor invasion[56,57].

We have previously introduced here the concept of ECM anisotropy. Geometry and topographic reconfiguration of the stroma has been seen to differ in malignancies, such as colorectal[65,175,198], breast[19,26,27], urogenital[66,67], and ovarian[176] cancer. Although tissue anisotropy has yet to be studied in the urological field, analyzing the geometry of the tissue and its anisotropy could provide additional information that could potentially be used as an indicator of increased tissue stiffness and unbalanced/altered ECM production during disease. We have also proposed novel ECM remodelers: EVs and microbiota. EVs have been recognized as key signaling mediators in regulating the tumor microenvironment by transferring several bioactive molecules involved in reprogramming and microenvironment

remodeling, tumor angiogenesis, and metastasis[109–111]. They have emerged as pivotal mediators of intercellular communications in local and distant microenvironments under patho/physiological conditions. EVs contain bioactive materials such as proteins, nucleic acids, lipids, and several MMPs[94]. These metalloproteinases are involved in altering the make-up of EVs either through the shedding of transmembrane proteins or by directly contributing to ECM remodeling. Although the nucleic acid and proteomic contents of EVs have been studied for their roles in the development of various diseases and tissue repair, information regarding the secretion and biological activities of EV-associated matrix-remodeling enzymes and their regulators is only just beginning to emerge. We have also introduced the microbiota as an emerging mediator of ECM remodeling. Every tissue in the human body has a defined microbiota associated with health, and the several bacteria can interact with the host cells, with which they are living in close proximity, through a variety of signals[115]. For example, bacteria may influence host cell behavior by modulating the host ECM by releasing different enzymes towards the tissue, like collagenases[122–126], LOXs[133,134], and others[140]. At the same time, it is plausible that changes in the host ECM linked to pathological states, such as tumors, fibrosis, or inflammation, can have an impact on the bacterial species associated with the tissue, leading to the expansion of some of them at the expense of others[141]. Although only a few studies have started to tackle this bidirectional interaction between host ECM and tissue-associated microbiota, future studies will shed light on the relevance of our hypotheses.

Even if measured using very different techniques, it has been reported in the literature that tissue mechanics change when cancer occurs[17–19,37,55,65], with this change generally being an increase in tissue stiffness. We have discussed that tissue mechanics differ in several urological-related pathologies, affecting different organs and tissues, such as the kidney, bladder, prostate, urethra, and testis. But, what are the mechanisms behind this change in the tissue mechanics? Is there a common mechanism affecting all different organs causing the increase in tissue stiffness? Or does a tissue-specific mechanism increase stiffness differently in each different tissue? Several general mechanisms causing an increase in tissue stiffness have been described: inflammation and infiltration of immune cells[16,26], remodeling of ECM, an increase in collagen deposition and linearization[19,23,175,176], and fibrotic change[66,67,164]. In addition, the role of specific mediators has been described in more detail for certain organs, such as Activin A for the kidney and testis[159,199], and collagen family members of the ECM-receptor interaction signaling pathway in BCa[66,174].

In the last years, systemic treatments with immunotherapy agents have shown promising results for the treatment of cancers, with the immunotherapy response being associated with a decrease in viable tumor cells and increases of immune content, which causes stromal and fibrosis activation due to effects on immune cell function[200]. As reported for several solid tumors, the stiffness of the liver increased in the case of hepatocellular carcinoma (HCC), from 3.2 to 5 kPa as evaluated by MRE[201]. In HCC it was found that HCC stiffness increases for MRE in patients treated with pembrolizumab, and the increased stiffness was significantly correlated with overall survival, time to progression, and the infiltration of immune cells[201]. Integration of the MRE in the diagnostic algorithm for GU malignancies for which immunotherapy has proven efficacious, such as kidney and bladder cancers, may aid in determining which patients would or would not benefit from immune checkpoint inhibitors, as well as help distinguish pseudo-progressions from bona fide progressions[202,203].

In the urological field, another well-known example of a mechanomedicine therapy comes from the treatment of Peyronie's disease (PD). PD is caused by the formation of an abnormal fibrotic plaque in the tunica albuginea of the penis, resulting in a curvature of the penile shaft that may impair sexual intercourse[204]. These fibrotic plaques are routinely evaluated by different ultrasound techniques, including SWE to evaluate tissue elasticity while detecting non-palpable plaques[205]. Caused by an excessive healing response, Peyronie's plaques are generated due to an increased synthesis of connective tissue and an inhibition of collagenase enzymes[206]. Among the potential treatments for this condition, the intralesional injection of collagenase purified from *Clostridium histolyticum* has been used since 1985, and results in a decrease in penile curvature in about one third of treated patients[207]. This collagenase degrades both collagen type I and II within the PD plaque[208].

Emerging information from the mechanomedicine field reveals the impact of the mechanical properties of tissues on disease onset and progression. The development of therapeutic strategies targeting tissue stiffness might have important clinical potential, not only because stromal stiffness is a hallmark of cancer that facilitates metastasis but also because it impedes the transport of therapeutic agents, reducing the efficacy of the treatment[26].

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

## Acknowledgements
This work has received funding from the European Union's Horizon 2020 research and innovation program under grant agreement no. 801126 (EDIT), and the European Union's support under the Marie Sklodwska-Curie grant agreement no. 812772 (Phys2BioMed). The Authors thank Dr. Dana Kuefner for her help with English grammar revision and insightful discussion.

## Author contributions
L.M.V. drafted the first version on the manuscript and collected inputs from all authors. V.M. edited the figures and contributed to writing the manuscript. F.P., C.V., and M.A. contributed to writing the manuscript and revised and edited the text. M.B., A.N., and A.S. provided a clinical view of the topic. All authors contributed to discussions and structure definitions of the respective sections.

## Competing interests
The authors declare no competing interests.
