## [Transparent Peer Review File · Communications Biology]

Reviewers' comments:
Reviewer #1 (Remarks to the Author):

[Comments are attached as a word doc, probably easier to read because of formatting]

[**Editor's note: the contents of the Word doc were used to make this file**]

Mechanics in urology: novel determinants and clinical relevance of tissue stiffness

In this review, Martinez-Vidal and colleagues review mechanical changes during urological disease, with a specific emphasis on stiffness caused by extracellular matrix (ECM) remodelling. I find the review timely and relevant, especially given the high interest in mechanobiology of diseases in the last few years. The authors describe how the ECM can be remodelled, including an interesting section on novel mediators (extracellular vesicles and the microbiota) which many, including myself, have likely not previously considered. They then describe data on stiffness values obtained during normal and diseased urological organs. My major comments, below, relate to including some extremely relevant areas which are not touched upon by the authors in order to put their manuscript into a better, fuller context for the reader.

Major Comments

In the introduction (page 4), I do not think the authors do justice to emphasising the relevance of their manuscript. The authors only mention a couple of examples: cancer and fibrosis, in passing, that has shown stiffness changes. But there are many, many others. Some examples are inflammatory diseases like atherosclerosis, other vascular diseases, aging, wounding, glioma, and neurodegenerative diseases like Alzheimer's & Parkinson's. I believe the authors should more strongly emphasise the wide relevance of mechanical changes in disease initiation and progression. They might want to detail specific examples and/or make a figure illustrating the wide range of diseases in which stiffness contributes, which I think would be very useful, novel and of general interest to the community. The authors should also mention that stiffness is only one mechanical parameter among many, although it is by far the best described one and therefore is the one focused on in the manuscript.

A very important general comment throughout the manuscript is that there are several sentences in which factual statements are made without any references. For example, on page 4 "cancerous cells...softer than normal cells," on page 31 "an interesting example on a murine mouse model..." or on page 6 "MMPS...increase matrix stiffness." For obvious reasons, I urge the authors to carefully check all factual statements contain references. In the case of the last example I mentioned, this immediately caught my attention because, to my knowledge, the prevailing view is the opposite of this. MMPs would degrade the matrix, thereby softening it, which fits with the general idea that matrix density roughly correlates with stiffness. If the authors believe their original statement true, they should provide conceptual reasoning and reference data that supports it.

Along a similar line to the above comment, the authors do not explicitly state what modulations the ECM can undergo which affects its stiffness. To non-experts, it is likely not obvious the many ways in which ECM can be modified to affect its mechanics. I strongly suggest bluntly listing and conceptually explaining this at the beginning of the section second (titled ECM). For example, mentioning that crosslinking, density, degradation, composition, fibre alignment and many other processes can all affect stiffness and providing a conceptual reasoning for each one. This would make the rest of the manuscript much easier to understand. It is also worth mentioning that the emergence of mechanics from a knowledge of the biological basis is extremely non-trivial because

rigidity is an emerging system level property that depends on both cellular/extracellular properties, their spatial architecture and interactions with the tissue.

The authors focus on the ECM's contribution to stiffness but do not mention the cellular contribution. It is clear that the manuscript is focusing mainly on ECM, but I believe a paragraph is warranted for the cellular contribution to stiffness, in order to give a full picture about where in vivo mechanics comes from. Cell density and cell contractility are both well known cellular parameters that directly affect tissue stiffness; not all the stiffness comes from the ECM which is the impression many will get from the manuscript in its current form. On a related point, the authors focus on enzymatic regulation of the ECM/stiffness but fail to mention that cellular behaviour in a completely physical way can alter the ECM and its stiffness. Authors should see work such as Glentis et al., Nat Comms 2017 from the Vignjevic lab, or Kelley et al., 2019 Dev Cell from the Sherwood lab for such data that clearly show physical changes (pores, ECM softening etc) of the ECM in MMP-independent conditions, which may explain why MMP inhibitors have failed in clinical trials. The authors should consider adding a section (or at least a couple of paragraphs) that explores the cells' contribution to stiffness in the ways described in this paragraph.

In section 5 onwards (page 16-), the authors describe data on the stiffness of different urological organs. Included in this description is a mention of a few different techniques in which such measurements were acquired (e.g. SWE, ultrasound) but how these techniques work in determining tissue stiffness is not explained. The authors should give at least a brief explanation about how these techniques work, and the advantages and disadvantages. Also, a non-specialist may assume that such techniques may give equivalent measurements on the same sample, which is rarely true, and one of the problems in mechanobiological measurements (mentioned by the authors in the conclusion). A figure may be a useful way to represent this.

Minor comments

Spelling and grammatical mistakes throughout, please check.

Page 5 – TGF- β modulates ECM – how? And how does this affect stiffness? Unclear/not stated in its current form.

Page 7 – the authors say “no clinical trials have started” but this is incorrect. MMP inhibitors were a major area of research and many clinical trials performed about 20 years ago but failed to progress (one potential reason described in my comments above).

Page 19 – What is TURBT? Not sure this is important information.

Fig. 1 – The text that is on drawing parts of the figure is almost impossible to read.

Reviewer #2 (Remarks to the Author):

Martinez-Vidal and colleagues present a review on urological pathologies from a “mechanomedicine” perspective. The primary discussion throughout the manuscript is around the concept of altered tissue stiffness with disease onset in urological tissues (kidney, bladder, prostate, and testis). The authors first focus on the classical mediators of ECM remodeling and tissue stiffness, such as stromal fibroblasts and macrophages, and then present more emerging regulators of tissue mechanics, the microbiome and extracellular vesicles secreted by a number of cancer and cancer-associated cells. In the final sections of the review the authors focus on tissue-specific mechanisms of altered mechanics with disease onset as well as the physiological consequences of such transformations.

Conceptually this review is timely and will be of interest to the community as it highlights the emerging role of the biophysical properties of the microenvironment, namely its stiffness, in promoting and driving urological pathologies. Overall, it covers a broad range of concepts and is clearly written. There are, however, a few issues that dampen my enthusiasm on the manuscript and would benefit from attention from the authors:

1. The authors focus solely on tissue stiffness rather than urological mechanics as is suggested by the title, the abstract, and the usage of the word “mechanomedicine”. The stiffness-focused angle should be better motivated and, importantly, the concept of viscoelasticity (and elasticity) explained clearly and at the onset of the manuscript, rather than mentioned briefly in the bladder section.
2. Throughout the manuscript the authors refer broadly to increased stiffness of the various urological tissues in disease without specifying which compartments and anatomical regions exhibited such altered stiffness. This is especially true for the kidney tissue stiffness section starting on page 15 – which part of the kidneys stiffens exactly in the different pathologies? These discussions are of critical value to understanding the molecular underpinnings of these mechanical transformations in the microenvironment.
3. Related to the previous point, there is a conceptual disconnect between the earlier sections of the review (and figure 1) focusing on molecular mechanisms driving altered stiffness (ECM remodeling ~micron scale) and the discussions around clinically-relevant technologies, which likely lack such fine resolution and can access bulk mechanical properties of tissues (would seem to be in mm scale and a result of a multitude of tissue changes with disease, not just stiffness) – a discussion on the resolution of these technique and their ability to access ECM stiffness per se should be discussed.
4. The authors dive into ECM remodelers section without properly setting up the stage and introducing the extracellular matrix and its components. The authors must introduce the concept of the extracellular environment as well as the major constituents of the extracellular matrix components in each urological tissue, especially those whose levels or architecture are modified with pathologies. Further, relating back to previous point, the authors should indicate which features of ECM changes are being detected in their table 1 and throughout their tissue specific stiffening sections as in its current form it is impossible to compare the different presented measurements.

Additional points

1. References are rather limited and omit many seminal works in the field by Valerie Weaver, Patricia Keely (discussions around ECM stiffness and malignant transformations/stiffening), Zena Werb (MMP-ECM discussions), Peter Friedl, M Olson, John Condeelis (fiber orientation driving migration discussion) labs, etc. Some references seem to be incorrectly placed within the text (example ref 16 at the end of page 7, as well as reference 54 on page 9)
2. Statement “by degrading the ECM, MMPs, among other consequences increase matrix stiffening and interstitial fluid pressure” is incorrect with respect to stiffness. Further, while MMPs can indeed modulate the ECM, ECM stiffness has also been shown to modulate MMP activity
3. Discussions about the relationship between tractions and migration (page 9) are over-simplified (as there are many modes of invasion/migration) as are discussion on page 5-6 regarding the relationship between extracellular matrix remodeling and invasion/metastasis since the ECM can be both pro-tumorigenic/pro-metastatic but also limit tumor cell invasion by serving a physical barrier to tumor cells.
4. The style of writing and referring to previously published work changes a few times throughout the manuscript (page 11 for instance)

5. Paragraph 2 and 3 on page 12 lack proper synthesis of information and do not read well
6. The section on microbiota-mediated modification in ECM properties and mechanics (especially pages 14-15) provide very intriguing hypotheses and it is surprising that these emerging mediators of ECM stiffness are not discussed again in conclusions
7. Discussion about heterogeneity of tissue mechanics ("from zone to zone") is made in the context of prostate cancer only and is also missing a reference (page 25). This is the case for every single tissue in the body and the reason why it is critical to precisely describe where each stiffness measurements have been made throughout the manuscript.
8. Sections on tissue-specific mechanisms for increased stiffness contain very little mechanistic literature, especially for bladder and prostate sections.
9. The usefulness of table 1 is not clear in the current form as units and techniques change – what is the main take home message?
10. Discussions about cell rheological changes and nuclear stiffness changes seems somewhat outside of the scope of this review and it is odd to discuss these just for one tissue (page 26; prostate section)
11. Conclusions section seems incoherent, instead of providing synthesis of the review and emerging concepts from it, it brings up totally new concepts such as matrix anisotropy, as well as somewhat over-generalizes the role of matrix mechanics in disease onset and progression, and discussions on limitations of mechanical measurements. To improve the manuscript, the conclusion section would benefit from extensive editing to synthesize the unifying factors with respect to mechanical changes in urological pathologies.

Below, the Reviewers' questions are reported in bold and the reply in Italic.

Reviewer #1 (Remarks to the Author):

Mechanics in urology: novel determinants and clinical relevance of tissue stiffness
In this review, Martinez-Vidal and colleagues review mechanical changes during urological disease, with a specific emphasis on stiffness caused by extracellular matrix (ECM) remodelling. I find the review timely and relevant, especially given the high interest in mechanobiology of diseases in the last few years. The authors describe how the ECM can be remodelled, including an interesting section on novel mediators (extracellular vesicles and the microbiota) which many, including myself, have likely not previously considered. They then describe data on stiffness values obtained during normal and diseased urological organs. My major comments, below, relate to including some extremely relevant areas which are not touched upon by the authors in order to put their manuscript into a better, fuller context for the reader.

We thank the Reviewer for appreciating the novel aspects of our review, its relevance and its timely nuance.

Major Comments

In the introduction (page 4), I do not think the authors do justice to emphasising the relevance of their manuscript. The authors only mention a couple of examples: cancer and fibrosis, in passing, that has shown stiffness changes. But there are many, many others. Some examples are inflammatory diseases like atherosclerosis, other vascular diseases, aging, wounding, glioma, and neurodegenerative diseases like Alzheimer's & Parkinson's. I believe the authors should more strongly emphasise the wide relevance of mechanical changes in disease initiation and progression. They might want to detail specific examples and/or make a figure illustrating the wide range of diseases in which stiffness contributes, which I think would be very useful, novel and of general interest to the community.

We thank the Reviewer for this comment. In order to emphasize the relevance of our manuscript, we elaborated a Figure illustrating how stiffness is altered in different organs of the human anatomy when disease, including neurological disorders, liver fibrosis, scarring and wound healing, etc.

The authors should also mention that stiffness is only one mechanical parameter among many, although it is by far the best described one and therefore is the one focused on in the manuscript.

We thank the reviewer for this comment. In order to make it more clear for the reader we added a box with the basic concepts of mechanics describing stiffness and viscoelasticity, and explain why we focus on tissue stiffness (Box 1), which is by far the best described mechanical parameter on tissues.

A very important general comment throughout the manuscript is that there are several sentences in which factual statements are made without any references. For example, on page 4 “cancerous cells...softer than normal cells,” on page 31 “an interesting example on a murine mouse model...” or on page 6 “MMPs...increase matrix stiffness.” For obvious reasons, I urge the authors to carefully check all factual statements contain references. In the case of the last example I mentioned, this immediately caught my attention because, to my knowledge, the prevailing view is the opposite of this. MMPs would degrade the matrix, thereby softening it, which fits with the general idea that matrix density roughly correlates with stiffness. If the authors believe their original statement true, they should provide conceptual reasoning and reference data that supports it.

We thank the reviewer for rising up this point, additional references have been added to the factual statements here discussed

Along a similar line to the above comment, the authors do not explicitly state what modulations the ECM can undergo which affects its stiffness. To non-experts, it is likely not obvious the many ways in

which ECM can be modified to affect its mechanics. I strongly suggest bluntly listing and conceptually explaining this at the beginning of the section second (titled ECM). For example, mentioning that crosslinking, density, degradation, composition, fibre alignment and many other processes can all affect stiffness and providing a conceptual reasoning for each one. This would make the rest of the manuscript much easier to understand.

We thank the reviewer for this comment. We added a box summarising main concepts of the ECM remodelling that will finally contribute to stiffness. Furthermore, on chapter 3 we better explained the matrix modifications that ultimately affect tissue stiffness.

Box 2. Main concepts on ECM remodeling contributing to stiffness

Desmoplasia: increased matrix production with remodeling of connective tissue structures adjacent to the tumor²¹⁸.

Cancer-associated fibroblasts (CAFs): the main source of ECM production and remodeling within the tumor microenvironment, promoting neoangiogenesis and epithelial to mesenchymal transformation (EMT)¹⁹.

Lysyl oxidase (LOX): enzyme that converts lysine into high reactive aldehydes forming crosslinking between fibers of collagens and elastin²¹⁹.

Metalloproteases (MMPs): extracellular enzymes that degrade the ECM components²²⁰.

Tumor-associated macrophages (TAMs): polarized macrophages that suppress antitumor immunity and promote tumor progression²²¹.

Anisotropy: reorganization of the topography of the ECM toward linearization of the alignment of ECM fibers²²².

It is also worth mentioning that the emergence of mechanics from a knowledge of the biological basis is extremely non-trivial because rigidity is an emerging system level property that depends on both cellular/extracellular properties, their spatial architecture and interactions with the tissue.

We thank the reviewer for pointing out this issue, and we agree that cellular contribution was missing in our previous manuscript. We introduced a new subchapter on cellular contribution to tissue stiffness. Furthermore, we better describe the crosstalk between cells and ECM and viceversa, for example describing the novel concept of durotaxis.

The authors focus on the ECM's contribution to stiffness but do not mention the cellular contribution. It is clear that the manuscript is focusing mainly on ECM, but I believe a paragraph is warranted for the cellular contribution to stiffness, in order to give a full picture about where in vivo mechanics comes from. Cell density and cell contractility are both well-known cellular parameters that directly affect tissue stiffness; not all the stiffness comes from the ECM which is the impression many will get from the manuscript in its current form.

As just mentioned above, we introduced a chapter on cellular contribution, including cell stiffness, density and traction. In addition we added a paragraph at the end of the introduction chapter pointing out this fact “Pathological changes in tissue stiffness can always be traced to altered amounts and/or function of its two fundamental constituents: cells (number and/or phenotype) and ECM (deposition and/or degradation).”

On a related point, the authors focus on enzymatic regulation of the ECM/stiffness but fail to mention that cellular behaviour in a completely physical way can alter the ECM and its stiffness. Authors should see work such as Glentis et al., Nat Comms 2017 from the Vignjevic lab, or Kelley et al., 2019 Dev Cell from the Sherwood lab for such data that clearly show physical changes (pores, ECM softening etc) of the ECM in MMP-independent conditions, which may explain why MMP inhibitors have failed in clinical trials. The authors should consider adding a section (or at least a couple of paragraphs) that explores the cells’ contribution to stiffness in the ways described in this paragraph.

We thank the reviewer for the very interesting bibliography and implemented such information on the chapter “Cellular contribution to stiffness”.

In section 5 onwards (page 16-), the authors describe data on the stiffness of different urological organs. Included in this description is a mention of a few different techniques in which such measurements were acquired (e.g. SWE, ultrasound) but how these techniques work in determining tissue stiffness is not explained. The authors should give at least a brief explanation about how these techniques work, and the advantages and disadvantages. Also, a non-specialist may assume that such techniques may give equivalent measurements on the same sample, which is rarely true, and one of the problems in mechanobiological measurements (mentioned by the authors in the conclusion). A figure may be a useful way to represent this.

We thank the reviewer for rising up this point. We included a Table adapted from Guimaraes et al, listing main techniques for the mechanical characterization of tissues and including the main techniques here discussed (RTE, SWE). In this table, we indicate the scale at which each different technique works, and the sample requirements (ex vivo or in vivo), showing how the techniques differ and provides an explanation why measurements by different techniques are not completely equivalent. Furthermore, at the beginning of chapter 5 we highlight “Before proceeding forward, it should be clear to the reader that the available different techniques to study tissue and organ stiffness vary in the resolution of the appreciable modifications”.

Minor comments

Spelling and grammatical mistakes throughout, please check.

Page 5 – TGF- β modulates ECM – how? And how does this affect stiffness? Unclear/not stated in its current form.

This has been clarified in chapter 2.2, and on 4.a (EVs): TGF- β (secreted by macrophages and contained in EVs), is cleaved by MMPs and promote angiogenesis

Page 7 – the authors say “no clinical trials have started” but this is incorrect. MMP inhibitors were a major area of research and many clinical trials performed about 20 years ago but failed to progress (one potential reason described in my comments above).

A paragraph at the end of the cellular contribution chapter (3.2) has been added.

Page 19 – What is TURBT? Not sure this is important information.

We agree with the reviewer, this information has been deleted

Fig. 1 – The text that is on drawing parts of the figure is almost impossible to read.

We have improved the figure and increased the size of the text.

Reviewer #2 (Remarks to the Author):

Martinez-Vidal and colleagues present a review on urological pathologies from a “mechanomedicine” perspective. The primary discussion throughout the manuscript is around the concept of altered tissue stiffness with disease onset in urological tissues (kidney, bladder, prostate, and testis). The authors first focus on the classical mediators of ECM remodeling and tissue stiffness, such as stromal fibroblasts and macrophages, and then present more emerging regulators of tissue mechanics, the microbiome and extracellular vesicles secreted by a number of cancer and cancer-associated cells. In the final sections of the review the authors focus on tissue-specific mechanisms of altered mechanics with disease onset as well as the physiological consequences of such transformations.

Conceptually this review is timely and will be of interest to the community as it highlights the emerging role of the biophysical properties of the microenvironment, namely its stiffness, in promoting and driving urological pathologies. Overall, it covers a broad range of concepts and is clearly written. There are, however, a few issues that dampen my enthusiasm on the manuscript and would benefit from attention from the authors.

We thank the Reviewer for acknowledging that our manuscript is both relevant and timely.

1. The authors focus solely on tissue stiffness rather than urological mechanics as is suggested by the title, the abstract, and the usage of the word “mechanomedicine”. The stiffness-focused angle should be better motivated and, importantly, the concept of viscoelasticity (and elasticity) explained clearly and at the onset of the manuscript, rather than mentioned briefly in the bladder section.

We thank the reviewer for this comment. We improved the title of our manuscript to “Causal contributors to tissue stiffness and clinical relevance in urology” to better emphasize the focus of the review. We also added a box on main concepts of mechanics to explain more clearly stiffness and elasticity (Box 1), and justify why we focus on tissue stiffness (Page 4, end of introduction chapter).

2. Throughout the manuscript the authors refer broadly to increased stiffness of the various urological tissues in disease without specifying which compartments and anatomical regions exhibited such altered stiffness. This is especially true for the kidney tissue stiffness section starting on page 15 – which part of the kidneys stiffens exactly in the different pathologies? These discussions are of critical value to understanding the molecular underpinnings of these mechanical transformations in the microenvironment.

We thank the reviewer for rising up this drawback of the techniques here discussed. As mentioned in the beginning of chapter 5 “From micro to macroscale: stiffness in clinical practice: imaging-based techniques, like MRE, SWE and RTE, can only provide macroscale maps of tissue stiffness, thus providing an overview of the organ stiffness”. In the case of the kidney, we do specify which part was used to localize the RTE, being the renal parenchyma for many of them (references 142-145, 152), renal cortex for reference 143, and cortical and corticomedullary stiffness values are given by references 148-150. Nevertheless, the elastography techniques here described have the main disadvantage of measuring stiffness at the macroscale and thus, makes it difficult to correlate it to specific anatomical compartments.

Furthermore, we included the discussion section “how to reconcile the contribution of cellular and microenvironment modifications to macroscale observations?” to highlight this gap of knowledge and unmet need in the field.

3. Related to the previous point, there is a conceptual disconnect between the earlier sections of the review (and figure 1) focusing on molecular mechanisms driving altered stiffness (ECM remodeling ~micron scale) and the discussions around clinically-relevant technologies, which likely lack such fine resolution and can access bulk mechanical properties of tissues (would seem to be in mm scale and a result of a multitude of tissue changes with disease, not just stiffness) – a discussion on the resolution of these technique and their ability to access ECM stiffness per se should be discussed.

We thank the reviewer for this critical comment. This technical difficulty is for sure an unmet need on the field, and specified at the beginning of the organ-specific chapter paper. Nevertheless we here aimed to describe the possible underlying mechanisms of altered stiffness when pathology at the molecular level (ECM remodelling, cell contributors, EVs and microbiome), even though the reported alterations more relevant on the clinics are performed at the macroscale tissue level as limitation of elastography techniques more widely used on the clinics. As stated above, aiming to reconcile these two topics we introduced the discussion chapter ““how to reconcile the contribution of cellular and microenvironment modifications to macroscale observations?”

4. The authors dive into ECM remodelers section without properly setting up the stage and introducing the extracellular matrix and its components. The authors must introduce the concept of the extracellular environment as well as the major constituents of the extracellular matrix components in each urological tissue, especially those whose levels or architecture are modified with pathologies.

We thank the reviewer for the suggestion and we included an introduction on ECM definition. We nevertheless did not include a description of the extracellular environment for each urological tissue, for the sake of the length of the review, and because we here refer to general ECM mechanisms that are valid for the ECMs of all organs.

Further, relating back to previous point, the authors should indicate which features of ECM changes are being detected in their table 1 and throughout their tissue specific stiffening sections as in its current form it is impossible to compare the different presented measurements.

This is actually one of the main difficulty on tissue mechanics: there is a lack of standardization between techniques, and the techniques presented in this review to calculate stiffness values of organs and diseases (elastography techniques reported on the table) are performed on the macroscale, as clarified at the beginning of organ-specific tissue stiffness chapter. As mentioned above on comment 2) and 3) at present it is not possible to correlate those values to specific features of ECM. We do introduce a paragraph discussing the correlation between microscale contributors to macroscale observations at the beginning of chapter 5, also a box clarifying ECM changes that can directly contribute to tissue stiffness and finally, we further discussed this gap of knowledge on the dedicated discussion chapter.

Additional points

1. References are rather limited and omit many seminal works in the field by Valerie Weaver, Patricia Keely (discussions around ECM stiffness and malignant transformations/stiffening), Zena Werb (MMP-ECM discussions), Peter Friedl, M Olson, John Condeelis (fiber orientation driving migration discussion) labs, etc. Some references seem to be incorrectly placed within the text (example ref 16 at the end of page 7, as well as reference 54 on page 9)

We thank the reviewer for the comment. References have been added.

2. Statement “by degrading the ECM, MMPs, among other consequences increase matrix stiffening and interstitial fluid pressure” is incorrect with respect to stiffness. Further, while MMPs can indeed modulate the ECM, ECM stiffness has also been shown to modulate MMP activity

We have amended the statement and better discussed that section.

3. Discussions about the relationship between tractions and migration (page 9) are over-simplified (as there are many modes of invasion/migration) as are discussion on page 5-6 regarding the relationship between extracellular matrix remodeling and invasion/metastasis since the ECM can be both pro-tumorigenic/pro-metastatic but also limit tumor cell invasion by serving a physical barrier to tumor cells.

Cell traction chapter has been included.

4. The style of writing and referring to previously published work changes a few times throughout the manuscript (page 11 for instance)

We thank the reviewer for rising up this concern, and we took care of improving the formatting and homogenize the way of citing published work.

5. Paragraph 2 and 3 on page 12 lack proper synthesis of information and do not read well

We have rephrased the paragraphs.

6. The section on microbiota-mediated modification in ECM properties and mechanics (especially pages 14-15) provide very intriguing hypotheses and it is surprising that these emerging mediators of ECM stiffness are not discussed again in conclusions

We thank the reviewer for the comment, we improved the conclusion section and added a paragraph summarizing the emerging ECM mediators on the conclusions.

7. Discussion about heterogeneity of tissue mechanics (“from zone to zone”) is made in the context of prostate cancer only and is also missing a reference (page 25). This is the case for every single tissue in the body and the reason why it is critical to precisely describe where each stiffness measurements have been made throughout the manuscript.

In the case of the kidney the reviewed literature do only specify cortex vs parenchyma, not for the rest of the organs. We replied to this comment on Major comments #2.

8. Sections on tissue-specific mechanisms for increased stiffness contain very little mechanistic literature, especially for bladder and prostate sections.

We thank the reviewer for rising up this concern. There is nevertheless not enough available literature on mechanistics, thus one of the purpose of the present review is to emphasize the relevant of stiffness and ECM remodelling processes behind it, and to point out the gap of knowledge on the responsible mechanisms. Furthermore, we included the discussion section “how to reconcile the contribution of cellular and microenvironment modifications to macroscale observations?” to highlight this gap of knowledge and unmet need in the field.

9. The usefulness of table 1 is not clear in the current form as units and techniques change – what is the main take home message?

We thank the reviewer for pointing this out. We added a paragraph mentioning differences between studies and the need for standardization, but still all studies do report that there are relative differences

and thus emphasize the potential of stiffness as a marker of disease. The main take home message is that, even though different techniques report different values and relative differences, the trend is the increase in tissue stiffness with progression of the disease.

10. Discussions about cell rheological changes and nuclear stiffness changes seems somewhat outside of the scope of this review and it is odd to discuss these just for one tissue (page 26; prostate section)

We agree with the reviewer with the incongruence of this information within this manuscript, therefore we removed this information on the revised manuscript.

11. Conclusions section seems incoherent, instead of providing synthesis of the review and emerging concepts from it, it brings up totally new concepts such as matrix anisotropy, as well as somewhat over-generalizes the role of matrix mechanics in disease onset and progression, and discussions on limitations of mechanical measurements. To improve the manuscript, the conclusion section would benefit from extensive editing to synthesize the unifying factors with respect to mechanical changes in urological pathologies.

We thank the reviewer for this criticism, we edited the conclusion chapter to first provide a summary of our main key concepts and take-home messages, and also included a critical Discussion chapter (“How to reconcile the contribution of cellular and microenvironment modifications to macroscale observations?”) recapitulating our main contributions/ideas to the field to highlight this gap of knowledge and unmet need in the field.

Reviewers' comments:

Reviewer #1 (Remarks to the Author):

The authors have addressed all of my comments, including new figures, a new chapter and correcting some incorrect statements. This review should be well received by the community.

Reviewer #2 (Remarks to the Author):

I maintain the this review is timely and would be of interest to the community. The authors have strengthened the manuscript conceptually and added a number of helpful tables/graphics since the first submission and addressed the majority of my comments, in principle. However, although potentially helpful and aiming to tackle many of the major criticisms, the "how to reconcile the contribution of cellular and microenvironment modifications to macroscale observations?" section is written in poor English and lacks proper discussion, depth, and references. Similarly, while conceptually-improved, the "conclusions" section, especially pages 30-32, is also poorly written and needs to be rewritten for clarity and in proper English with correct grammar. Further, referencing is also extremely thin in this section and concepts are very vague, case in point the last paragraph of the review: lines 759-765 read "One of the most important questions that the mechanomedicine field needs to address is: since stromal stiffening is related with poor prognosis, can we revert or block further matrix remodeling to prevent metastasis (ref 26)

An interesting example on a murine model showed an increase in tumor stiffness with disease progression, and after treatment the tumor became softer and smaller. Development of therapeutic strategies targeting tissue stiffness has an important clinical potential. Not only because stromal stiffness is a hallmark of cancer that facilitates metastasis, but it also impedes transport of therapeutic agents, reducing the efficacy of the treatment (ref 26)."

First of all there is a huge body of primary literature that needs to be referenced around discussion of these concepts versus a small mini-review. Secondly, there is a very strange discussion around some unreferenced murine model. This isn't an effective or convincing way to communicate concluding concepts of the review.

Below, the Reviewers' questions are reported in bold and the reply in Italic.

Reviewer #1 (Remarks to the Author):

The authors have addressed all of my comments, including new figures, a new chapter and correcting some incorrect statements. This review should be well received by the community.

We thank the Reviewer for appreciating the improvement of our manuscript and acknowledging that it would be of interest to the community.

Reviewer #2 (Remarks to the Author):

I maintain the this review is timely and would be of interest to the community. The authors have strengthened the manuscript conceptually and added a number of helpful tables/graphics since the first submission and addressed the majority of my comments, in principle. However, although potentially helpful and aiming to tackle many of the major criticisms, the “how to reconcile the contribution of cellular and microenvironment modifications to macroscale observations?” section is written in poor English and lacks proper discussion, depth, and references.

We thank the Reviewer for appreciating the refinement of our manuscript, the implementation of their suggestions and comments and the enhancement of its content by including the new figures. We corrected the English grammar and writing style in the whole manuscript, especially in the “how to reconcile the contribution of cellular and microenvironment modifications to macroscale observations?” chapter with the help of a native English speaker. We included additional references on this chapter and improved its discussion.

Similarly, while conceptually-improved, the "conclusions" section, especially pages 30-32, is also poorly written and needs to be rewritten for clarity and in proper English with correct grammar. Further, referencing is also extremely thin in this section and concepts are very vague, case in point the last paragraph of the review: lines 759-765 read "One of the most important questions that the mechanomedicine field needs to address is: since stromal stiffening is related with poor prognosis, can we revert or block further matrix remodeling to prevent metastasis (ref 26) An interesting example on a murine model showed an increase in tumor stiffness with disease progression, and after treatment the tumor became softer and smaller. Development of therapeutic strategies targeting tissue stiffness has an important clinical potential. Not only because stromal stiffness is a hallmark of cancer that facilitates metastasis, but it also impedes transport of therapeutic agents, reducing the efficacy of the treatment (ref 26)."

First of all there is a huge body of primary literature that needs to be referenced around discussion of these concepts versus a small mini-review. Secondly, there is a very strange discussion around some unreferenced murine model. This isn't an effective or convincing way to communicate concluding concepts of the review.

We thank the Reviewer for pointing out this drawback of our conclusion chapter. We have improved the English grammar and added the missing references to back up the statements here made. We have removed the discussion about the murine model at the end of the manuscript and added a final improved paragraph on the conclusion aiming to better communicate the clinical importance of tissue stiffness and its potential to be used as a treatment target.